# Comparison of the Physical Demands of Friendly Matches and Different Types On-Field Integrated Training Sessions in Professional Soccer Players

**DOI:** 10.3390/ijerph17082904

**Published:** 2020-04-22

**Authors:** Jesus Vicente Giménez, Julen Castellano, Patrycja Lipinska, Mariusz Zasada, Miguel-Ángel Gómez

**Affiliations:** 1School of Sport Sciences, Universidad Europea de Madrid, 28670 Villaviciosa de Odón (Madrid), Spain; 2International University of Valencia (VIU), 46002 Valencia, Spain; 3Department of Physical Education and Sport, University of the Basque Country (UPV/EHU), Vitoria, 48940 Leioa, Spain; julen.castellano@ehu.eus; 4Institute of Physical Education, Kazimierz Wielki University, 85-064 Bydgoszcz, Poland; patlipka@gmail.com (P.L.); mariusz.zasada@ukw.edu.pl (M.Z.); 5Faculty of Physical Activity and Sport Sciences, Technical University of Madrid, 28040 Madrid, Spain; miguelangel.gomez.ruano@upm.es

**Keywords:** task performance, athletic performance, analysis, soccer

## Abstract

The aim of this study was to investigate the relationships among physical demands of two friendly matches (FMs) and three task training sessions (TS_1,2,3_) combining in a different way: a Small-Sided Game (SSG), Mini-Goals (MG), a ball Circuit Training (CT) and a Large-Sided Game (LSG): SSG+MG+LSG (TS_1_), SSG+CT+LSG (TS_2_) and MG+CT+LSG (TS_3_). The TS and match demands in running intensities were monitored in fourteen professional soccer players (age = 23.2 ± 2.7 years, height = 178 ± 6 cm, body mass = 73.2 ± 6.9 kg, mean and SD, respectively) using 10-Hz global positioning system devices, and players’ perception of exertion was recorded after each session or match using a visual analogue scale. A one-way repeated measures ANOVA with a Bonferroni correction coupled with magnitude-based inferences were used. A principal component (PC) analysis was conducted on all variables to account for covariance. Three PCs were retained, explaining 76% of the variance: Component 1 explained 46.9% with the associated variables: Total Distance (TD) and distance covered in ranges of speed from >2.2 to <5 m/s, Player Load and Work Rest Ratio; component 2 explained 19.7% and was composed of TD at > 5 m/s and maximal running speed (MRS); and component 3 explained 9.5% and was represented by TD < 2.2 m/s, decelerations and accelerations. The ANOVA results showed significant differences (*p* < 0.05) among TS vs. FM in TD3, TD4, TD5, and TD > 5, TD, deceleration rate, acceleration rate, maximal running speed, exertion index, work rest ratio, and self-reported exertion. Therefore, the training routines did not replicate the main set of high intensity efforts experienced in competitive conditions. Additionally, PC analysis could be applied in order to select the most representative training and competitive conditions.

## 1. Introduction

The available research describes soccer as a unique world owing to its particular characteristics [1]. In fact, competitive soccer performance during match-play and training requires players be able to dominate technical and tactical skills of the game under high physical stressors [2]. Therefore, soccer players need to achieve high physical-conditioning levels to afford specific training adaptations [3]. Indeed, researchers have attempted to bridge the gap between theory and practice by developing specific training routines on technical, physical and tactical aspects during competitive situations [4]. Nowadays, the available literature has not studied which is the ideal routine in training sessions in order to replicate the physical demands of match-competitions.

A major interest in sports sciences has been the development of training programs that provide coaching staff with reliable methods for improving training while enhancing player’s performance [5]. Hence, several methods of specific training related to this sport have been established. For instance, exercises without a ball which develop physical capacities in isolation and new methods which simultaneously improve physical capacities along with technical and tactical skills in accordance with the contemporary physical demands of the match-play [6]. In effect, various reliable methods have been used to improve physical demands, including traditional conditioning training, traditional sprinting training, high intensity interval training (HIIT) and small-sided games (SSG), which are applied and tested in order to enhance the physical performance in soccer players [7]. Thus, this corroborated the literature that SSGs enable the development of both physical/physiological and technical/tactical skills at the same time, summarizing that SSGs are more effective than the traditional methods [8].

Consequently, despite our limited understanding of the dose–response relationship, there is interest by the sport science community for characterizing training protocols to bridge the gap between the design of training tasks and competition performance [9]. In order to do so, the training sessions should be tailored to replicate as close as possible the external and internal loads performed during match-play [10]. Additionally, previous literature suggests that performances may improve when training simulates and affords the physical and movement demands of competitive matches [11]. Consequently, the target to be achieved by the coaches and science staff is to implement stimuli and specific movement patterns associated with the match in workout sessions [12]. In fact, the lack of suitability of training loads (TLs) applied during training sessions compared to the physical demands of actual competitive soccer (matches) are related to the risk of non-contact injuries causing a reduction in performance and fitness in soccer players [13]. Currently, global positioning systems (GPS) are widely used in soccer training and permit valid and reliable estimates of the external load incurred during SSGs and FMs [14].

In this regard, training responses in athletes are generally related to the training stimuli (e.g., relative/internal training load) during the different training cycles [15]. Therefore, it is of paramount importance to monitor athletes’ fatigue, fitness and performance responses to the various training phases to adjust and individualize match training load (and contents) both during and between each training cycle [16].

In particular, little is known about some key components of physical demands (e.g., high-intensity efforts) assessed in drills of different types and sizes to stimulate external load in the same way as competitive soccer. This may be due to the unpredictable and multifactorial nature of soccer, involving great complexity in the quantification of the workouts. Thus, improving physical fitness during the in-season requires a fine-tuning of loading and unloading within the micro-cycle and increased training emphasis on multiple goals both within and between consecutive micro-cycles [17]. However, to the best of our knowledge no study has yet adopted such an approach to compare the interaction of different components assessed in the training sessions in professional soccer players. Therefore, the present research could provide in-depth information about the consistency and knowledge of physical demand components by comparing matches and training performance.

According to this rationale, the purpose of this study was to compare the relationships among physical demand indicators of professional soccer training during three types of TS and competitive soccer (FMs) in professional soccer players. It was hypothesized that when the players are training with SSGs they will perform higher intensity physical activity than during competitive match play (FMs).

## 2. Materials and Methods 

### 2.1. Experimental Design

An experimental randomized controlled trial was used to verify the differences among physical demands of two friendly matches (FMs) and three task training sessions (TS1,2,3). Players were previously familiar with the different task formats and the material used (GPS technology). The study was conducted for a 5-week period in the middle of a 9-month competitive season, from February to March in 2015. All training sessions were done once per week and the same day of the micro-cycle (i.e., Wednesday), and at the same time during the day (10:30–12:30), with two days of rest after the team’s official match and after a recovery session to avoid the onset of fatigue. The players were distributed into two teams based on skill level and playing position to balance the competitive level under criterium of the head coach. The teams did not permitted changes during the research plan.

### 2.2. Participants

The experiment population consisted of a convenience sample from a professional soccer team. Fourteen professional outfield male soccer players (mean ± SD, age = 23.2 ± 2.7 yr, height = 178 ± 6 cm, body mass = 73.2 ± 6.9 kg, body fat = 12.6 ± 2%, and soccer experience = 14 ± 5 yr) were recruited as volunteer subjects and were fully informed of any risks and discomforts associated with the study.

To ensure a homogeneous sample, only the players who played regularly in the official league matches were considered for the study (i.e., the criteria for inclusion was that the player must have played more than 65 min of total playing time during a regular match, trained six times per week during ≈ 1.5 h per session, and had not reported any injuries during the 3 months prior to the initial testing sessions). The participants provided their informed written consent to participate before starting field testing in accordance with the requirements of the Declaration of Helsinki (2013); the university ethics committee of the Faculty of Physical Activity and Sport Sciences Polytechnic University of Madrid (EP1004/2015) approved this study. The research also received formal approval from the Football Club of Concordia Elblag (II league, East, Poland). To ensure the confidentiality of the players, all performance data were anonymized before analysis.

### 2.3. Procedures

The trials were conducted three days after the previous official match and at the same time of day in order to limit the effects of fatigue and/or circadian variation. The same contents were organized in three different ways. The first experimental approach of the TS_(S1)_ included a Small-Sided Game (SSG), Mini-Goal games (MG) and a Large-Sided Game (LSG); the second TS_(S2)_ consisted of a SSG, CT and LSG; and the third TS_(S3)_ was designed with MG, CT and a LSG. The CT, SSGs, and MGs consisted of four repetitions of 4-min game play (the total playing time required during each task was 16 min) interspersed by 2 min of active recovery, and a LSG with eight players per side (total playing time required during each task was 32 min). The data obtained during the sessions were compared with the first 32-min periods of two friendly matches (FM) against a similarly ranked opponent (i.e., based on the league match by match ranking). The use of the same time duration (TS_s_ vs. FM, 64 min) for all game conditions ensured that each player’s performance was compared equally. All the training drills could be identified as one of the following training modes (Table 1 and Figure 1):

All the SSGs, LSGs and CT repetitions were played and completed by the same 14 players, and the game situations were performed at the club facilities on an outdoor artificial grass pitch with regular goals. Participants wore training clothes and soccer boots. In the SSGs, CT, LSGs and FM there was one goalkeeper per side, but their movements and actions were limited to saving the ball when a shot was aimed at their goal with no other movements allowed. Specific sub-components of each session were categorized according to the focus of training to evaluate the weekly organization of the training sessions and to provide physical demand data on particular aspects of training. This categorization was established following discussions with the team coaches, then the physical requirements and the training of random conditional fields were scheduled as a session devised to enable players to cope with the physical demands of match play [18].

### 2.4. Data Collection

The physical responses and time-motion characteristics of the players were monitored using Global Position System devices (GPS MinimaxX v4.0, Catapult Innovations, Melbourne, 144 Australia) operating at a sampling frequency of 10 Hz and incorporating a 100 Hz triaxial accelerometer. This technology has been previously validated and has proven reliable for monitoring soccer players’ movements and activities of different intensities [19]. In the present investigation, all data analyses were performed following the manufacturer’s specialized software package (Catapult Sprint version 5.0.9.2, Canberra, Australia), and before starting tests the software was updated according to the manufacturer’s recommendations, trying to avoid, as much as possible, variations in acceleration and deceleration measures [20], and to ensure that all parameters downloaded from the GPS devices provided reliable results and powerful findings [21]. As in previous studies [22], the total running distance covered was determined for each player according to five zones of increasing speed: walking (<2.2 m/s, TD2), jogging (from >2.2 to ≤3.3 m/s, TD3), low speed running (from >3.3 to ≤4.2 m/s, TD4), moderate speed running (from >4.2 to ≤5.0 m/s, TD5), and high-speed running (>5.0 m/s, TD > 5). The accelerations and decelerations were also measured with ±2 m/s^2^ intervals using the data provided by the GPS [23]. The work-to-rest ratio (W:R) was calculated as the distance covered by the player at a speed of >2.2 m/s (period of work, TD3 + TD4 + TD5 + TD > 5) divided by the distance covered at a speed of 0–2.2 m/s (period of recovery or rest, TD2). The exertion index (EI) was also calculated to identify the player’s cumulative physical load. This variable was derived from individual movement speeds and was calculated using the sum of the weighted instantaneous speed, the weighted accumulated speed over 10 s, and the weighted accumulated speed over 60 s. A further indicator used was player load (PL) obtained via accelerometry [24], combining the accelerations produced in the three planes of body movement by means of a 100 Hz triaxial accelerometer. W:R, EI and PL were assessed in arbitrary units (AU).

The day before the experimental trials, participants were instructed to avoid caffeine-containing products and the technical staff programmed a low-intensity, low-volume training session. The diet was standardized for the 24 h before the experimental trials (all the players lived in the same residence) and compliance was verified using self-report diaries. The day of the experiments, participants arrived at their habitual training facility at 10:30 a.m. and body mass (±0.05 kg) and body height (±0.1 cm) were measured (SECA285, Hamburg, Germany). Body fat percentage was estimated using segmental bioimpedance (BC-418-MA, Tanita, Tokio, Japan) following recommended standardizations for this measurement. Then, participants dressed in their competition clothes and a GPS device was provided for each player. Each training session began with a 25 min standard warm-up (running, stretching, and contact with the ball), followed by different drills (small-sided games, running exercises, technical and tactical drills). During the period before this study, no strength-training sessions were performed by the players. In order to avoid stoppage time, several balls were placed around playing areas for immediate availability. Two assistant coaches acted as timers and referees for the SSGs, LSGs and CT, while a professional referee was used for the FM. Just after the end of the game, participants individually gave their Rating of Perceived Exertion (RPE) using a modification of the method proposed by Borg (assessed in AU) [25].

### 2.5. Statistical Analysis

First, to check the differences between TS_(S)_ and FM, all the variables were initially checked for normality using the Shapiro–Wilk test and all of them presented a normal distribution (*p* > 0.05). Secondly, the four experimental situations TS_1(SSG+MG+LSG)_, TS_2(SSG+CT+LSG)_, TS_3(MG+CT+LSG)_ and the 2 FMs were compared using a one-way repeated measures ANOVA. When the ANOVA test showed a significant group effect, the Bonferroni post-hoc test (pairwise comparisons) was used. Effect size (ES) values were estimated using Cohen’s d with the following criteria: >0.2 (small); >0.6 (moderate); and >1.2 (large) [26]. Thirdly, an unsupervised model of machine learning for continuous variables was used to find the strongest features for each training task [27]. Then, the factor analysis using principal components (PCA) was performed on the physical indicators to reduce the dimensions of the variables (varimax rotation). The Kaiser-Meyer-Olkin measure was adequate (0.65). The PCA model obtained accounted for 76.2% of the total variance. Three factors were extracted with eigenvalues above 1, and the criterion of 0.70 was used to identify substantial loadings. The extracted factor scores were saved as variables to be compared in plotted graphs showing factor 1 vs. factor 2, factor 1 vs. factor 3 and factor 2 vs. factor 3. The data were analyzed with the IBM SPSS for Windows statistical package version 20.0 (IBM Corp., Armonk, NY) and the significance level was set at *p* < 0.05.

## 3. Results

The descriptive results for each variable according to TS and FM are presented in Table 2.

The results showed significant differences among TS vs. FM in TD3 (F_3,39_ = 13.10; *p* = 0.001), TD4 (F_3,39_ = 14.39; *p* = 0.002), TD5 (F_3,39_ = 9.63; *p* = 0.003), and TD > 5 (F_3,39_ = 21.15; *p* = 0.001), TD (F_3,39_ = 19.27), deceleration rate (F_3,39_ = 37.05; *p* = 0.001), acceleration rate (F_3,39_ = 41.70; *p* = 0.001), maximal running speed (F_3,39_ = 13.29; *p* = 0.001), exertion index (F_3,39_ = 10.51; *p* = 0.001), work rest ratio (F_3,39_ = 44.82; *p* = 0.001), and self-reported exertion (F_3,39_ = 25.68; *p* = 0.001).

The pairwise comparisons showed significant differences between FM vs. TS1 with more TD4 (*p* = 0.002; ES =−1.56), TD > 5 (*p* < 0.001; ES = −2.51), and TD (*p* < 0.001; ES = −1.71) during FM; lower deceleration rate (*p* < 0.001; ES = 2.91) and acceleration rate (*p* < 0.001; ES = 3.90) during FM; and higher MRS (*p* = 0.01; ES = −1.45), lower exertion index (*p* = 0.01; ES = 1.79) and self-reported exertion (*p* < 0.001; ES = 2.50) during FM.

The differences between FM vs. TS2 and TS3 showed significant differences with higher values for distances covered during FM at TD3 (*p* < 0.001; ES = −2.25 and −1.43, respectively), TD4 (*p* < 0.001; ES = −2.12 and −1.59, respectively), TD5 (*p* < 0.05; ES = −1.29 and −1.68, respectively), TD > 5 (*p* < 0.001; ES = −1.60 and −1.98, respectively), and TD (*p* < 0.001; ES = −2.70 and −2.16, respectively). TS2 vs. TS3 showed a higher number of accelerations (*p* < 0.001; ES = 1.61 and 2.85, respectively) than FM. In addition, the FM format showed higher MRS (p < 0.001; ES = −1.69 and −1.27, respectively) and work rest ratio (*p* < 0.05; ES = −5.0 and −1.20, respectively) and less self-reported exertion (*p* < 0.001; ES = 2.55 and 1.82, respectively) than TS2 vs. TS3, and a lower exertion index than session 3 (*p* = 0.01; ES = 1.89).

The differences among TS1, TS2 and TS3 were significant for TD3 between TS1 vs. TS2 (*p* = 0.001; ES = 1.29) and TD > 5 between TS1 vs. TS3 (*p* = 0.001; ES = −1.51); for decelerations between TS1 and TS2 (*p* < 0.001; ES = 3.39) and TS1 vs. TS3 (*p* < 0.001; ES = 3.40); for accelerations between TS1 vs. TS2, TS1 vs. TS3, and TS2 vs. TS3 (*p* < 0.05; ES = 2.20, 1.38, and −1.03, respectively); for work rest ratio between TS1 vs. TS2, and TS2 vs. TS3 (*p* < 0.001; ES = 4.00 and −4.67, respectively), and for self-reported exertion between TS1 vs. TS3, and TS2 vs. TS3 (*p* < 0.001; ES = 1.43 and 1.33, respectively).

The results of the PCA are shown in Table 3 and plotted in Figure 2. This analysis showed that there were three principal components (factors) that explained 76.2% of the total variance. The first component accounted for 46.9% of the variance and included the variables: TD, TD4, TD5, PL and work rest ratio. Component 2 accounted for 19.7% of the variance and included TD > 5 and MRS; and, finally, component 3 accounted for 9.6% of the variance and included TD2, decelerations and accelerations (number).

## 4. Discussion

The aim of the present study was to quantify the physical demands required by professional soccer players during TS and compare with FM. The current study (i) explored the soft assembly of physical demand patterns during a FM and three different TS typically prescribed during a standard week (with a game every seven days) by coaches in a professional football team, and (ii) also aimed to clarify the interactive effect of some indicators of movement patterns associated with a typical training session with several different components. The results showed intriguing findings that are not close to the competitive scenarios and reflect different levels of physical and physiological requirements of training associated with actual competition demands. This fact may suggest the need for specific designing of applied tasks related to match-play situations.

Thus, the comparisons among the three experimental TS and FMs were performed using daily workouts calculated as the sum of all training loads for specific training sessions performed in one day [28]. Even though the available research on the internal load associated with different types of training session components is very limited in professional male soccer players [19], our findings seem to confirm that TS were characterized by different demands in relation to the FM. For example, when the player trains with SSGs, MGs and LSGs, the session requires physically demanding accelerations and decelerations. These findings are in accordance with a previous study [29] where the logical use of the former type of SSG was justified to target development of power-related soccer actions. Our results are in line with previous studies [30], demonstrating that the number of accelerations was higher during SSGs used as part of training than it was during actual matches. This finding could be related to greater neuromuscular fatigue and increased metabolic cost during matches through overstimulation. This may be owing to inadequate applications of SSGs which develop more accelerations and decelerations. However, when the players played the FM, there were higher maximal running speeds (m/s) and different running peaks than in the session types studied. According to the available research [31], the high-intensity activity was suggested to be a better measure of physical performance during a soccer match. These variables seem to reflect the higher level of external work performed at high speed during the FMs than in the analyzed TSs. The available literature suggests that players require aptitudes for high intensity running [30]; unfortunately SSGs during training do not stimulate the high-intensity, repeated-sprint demands of FMs. These findings suggest that SSGs should be supplemented with game-specific training that simulates the high-intensity repeated-sprint demands of official matches. 

The findings of the current study are in contrast to others [32] as we cannot conclude that developments in physical training have important implications for the success of soccer players. It is because we did not find (when including CT) any superiority in typical soccer performances such as changes of direction, lateral running, sprints and shots on target in relation to MRS and TD5 developed in FMs. On the other hand, it was found [33] that by including CT, the players’ physical performance was more similar to real match competition (i.e., imposing the most critical demands such as higher maximal running speeds (m/s) and different running peaks). We also found this finding; however, there were significant effects among FM, SSGs and MGs for TD covered at >2.2 ≤3.3 m/s; >3.3 ≤4.2 m/s; >4.2 ≤5 m/s, TD, PL and W:R. These types of abilities are crucial for matches and may be used when coaches want to develop specific training models for competitive match play. 

In summary, the training sessions analyzed in the current study do not achieve the physical demands of competitive matches in the vast majority of physical performance variables. This fact suggests the need to be supplemented with high-intensity repeated sprint tasks as well as with the inclusion of another types of game formats.

Finally, one of the limitations of this study concerns the sample studied; it would be interesting to extend this study to include more participants and different categories and levels. Secondly, future studies using small sample sizes should consider the use of Bayesian models to check the differences among training tasks or players. Thirdly, task constraints during SSGs margins of victory (i.e., goal differences between teams), match/game status (winning, drawing or losing), and the order in which the exercises were performed were not accounted for in the present study and it would also be interesting for future studies to analyze these aspects. Therefore, prospective research in this area is necessary for a better understanding the main components of training sessions and their relationship with real scenarios in actual competition.

## 5. Conclusions

The findings of this study suggest that the training routines did not replicate the leading set of high-intensity efforts usually found in competitive conditions. In addition, the use of PCA analysis might bridge the gap between training routines and competitions since it has been able to identify physical demand relationships in competition and different components of the training session game play scenarios. From these data, the correct application of SSGs in relation to the purpose during the training process will achieve performance enhancement in both the soccer players’ and team’s behavior in order to complement or compensate for some performance deterioration. Overall, this result demonstrates that SSGs, MGs and LSGs demand a greater number of accelerations and decelerations and can be used as an effective training mode to enhance high intensity demands in soccer players. As far as we understand, the use of CT together with SSGs and LSGs allows for the development of generic aerobic training models. Finally, in order to improve the players’ performance according to match demands, the absolute space of the games should be increased and involve a greater number of players to replicate the exact movement patterns of competition.

## Figures and Tables

**Figure 1 ijerph-17-02904-f001:**
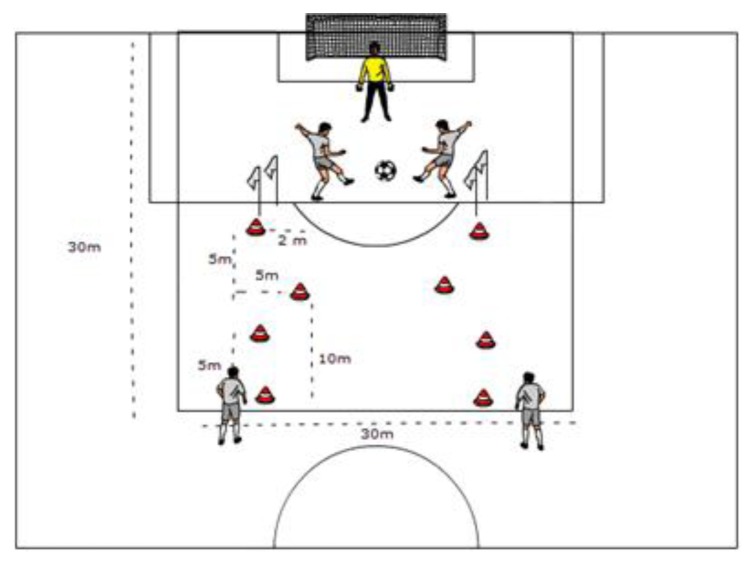
Circuit Training (i.e., 30 × 30 m^2^, individual occupied area per player = 90 m^2^).

**Figure 2 ijerph-17-02904-f002:**
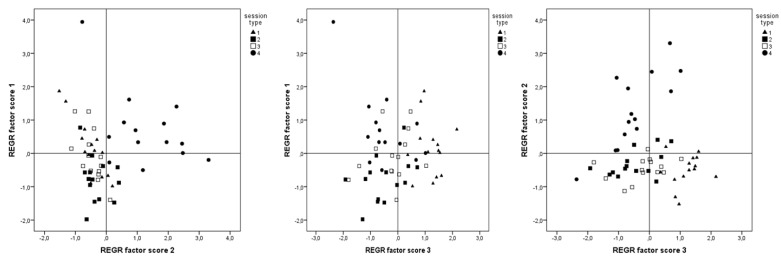
The variables grouped into three components. Note: The variables were grouped into three components. Only variables with values > 0.7 were considered. Component 1 involved Distance Covered in Different Range of Speed (from >2.2 to ≤3.3 m/s, from >3.3 to ≤4.2 m/s and >4.2 ≤5 m/s), Distance Total, Player Load and Work Rest Ratio (A.U). For Component 2, the variables were Distance Total > 5m/s and Maximal Running Speed (m/s); Component 3 included Distance Total in Speed Running <2.2 m/s, Decelerations Rate (number) and Accelerations Rate (number).

**Table 1 ijerph-17-02904-t001:** On-field integrated training routines included.

Task (Players)	Goalkeeper	Maximal Ball Touches Per Player	Field Dimension (m)	Field Area (m^2^)	Field Area Per Player (m^2^)	Training Prescription
SSG (4 vs. 4)	Present	2	30 × 30	900	112.5 m^2^	8 × 4-min + 2-
MG (4 vs. 4)	Absent	2	30 × 24	720	90 m^2^	8 × 4-min + 2-
CT (1 vs. 1)	Present	Unlimited	30 × 30	900	-	8 × 4-min + 2-
LSG (8 vs. 8)	Present	Unlimited	42.4 × 42.4	1798	112.38 m^2^	32 min
FM (11 vs. 11)	Present	Unlimited	90 × 70	6.300	308.75 m^2^	32 min

SSG = small-sided game; MG = mini goal game; CT = circuit training; LSG = large-sided game; FM = friendly match.

**Table 2 ijerph-17-02904-t002:** The descriptive results for each variable according to task training sessions (TS) and friendly matches (FM) are presented in Table 2. Data are mean ± SD for 14 players.

Variable	Session(1)	Sessión (2)	Sessson (3)	Session (4)
	SSG+MG+LSS	SSG+CT+LSS	MG+CT+LSS	F.M
Distance covered rate (m)	M ± SD	M ± SD	M ± SD	M ± SD
<2.2 m/s	3553.8 ± 191.4	3349.3 ± 189.5	3405.7 ± 178.9	3267.3 ± 288.2
>2.2 ≤3.3 m/s	1749.4 ± 376.6	1322.6 ± 286.5	1582.1 ± 338.7	2139.2 ± 438.2
>3.3 ≤4.2 m/s	667.6 ± 178.4	538.9 ± 154.5	663.0 ± 174.4	1100.7 ± 376.3
>4.2 ≤5 m/s	310.6 ± 132.1	382.0 ± 95.1	334.3 ± 81.3	578.0 ± 208.0
>5 m/s	124.6 ± 70.2	230.0 ± 109.4	219.1 ± 54.7	499.4 ± 228.2
Total	6405.4 ± 685.1	5822.7 ± 610.0	6204.1 ± 585.1	7584.5 ± 694
Decelerations rate (number)				
Total	920.1 ± 24.9	830.0 ± 28.2	804.8 ± 43.0	840.7 ± 29.6
Accelerations rate (number)				
Total	1425.6 ± 50.2	1310.8 ± 54.4	1361.2 ± 43.0	1224.1 ± 53.2
Indicators of workload				
Player Load	676.1 ± 77.0	609.0 ± 89.1	634.9 ± 96.1	703.0 ± 97.9
Maximal runing speed (m/s)	6.4 ± 0.3	6.1 ± 0.5	6.5 ± 0.3	7.2 ± 0.8
Exertion Index	49.9 ± 8.9	42.0 ± 8.0	50.1 ± 8.3	35.9 ± 6.7
Wor Rest Ratio (A.U)	0.8 ± 0.2	0.2 ± 0.1	0.9 ± 0.2	1.2 ± 0.3
Self-reported Exertion (A.U)	7.2 ± 0.4	7.1 ± 0.3	6.7 ± 0.3	5.7 ± 0.8

SSG = small-sided game; MG = mini goal game; LSS= large-side game; CT = circuit training; FM = friendly match; MD = mean differences; SD = standard deviation.

**Table 3 ijerph-17-02904-t003:** Component analysis values.

	Components											
Variables	1	2	3	4	5	6	7	8	9	10	11	12
Eigenvalue	6.098	2.564	1.239	0.769	0.682	0.488	0.415	0.339	0.16	0.1	0.079	0.066
% of Variance	46.91	19.72	9.53	5.92	5.25	3.76	3.19	2.61	1.23	0.77	0.61	0.51
% accumulated	46.91	66.63	76.16	82.08	87.33	91.08	94.28	96.88	98.11	98.88	99.49	100.
TD1	−0.28	−0.085	0.758 *									
TD2	0.905 *	0.059	−0.042									
TD3	0.896 *	0.232	−0.244									
TD4	0.753 *	0.291	−0.4									
TD5	0.436	0.734 *	−0.233									
TD	0.919 *	0.293	−0.034									
DEC (*n*)	0.134	−0.003	0.767 *									
ACC (*n*)	−0.023	−0.532	0.701 *									
PL(AU)	0.745 *	0.132	0.265									
MRS (m/s)	0.327	0.794 *	0.173									
W:R (AU)	0.849 *	0.212	−0.043									
EI (AU)	0.526	−0.615	0.301									

Note: TD1 2.2 ≤3.3 m/s; TD2 is >3.3 ≤4.2 m/s; TD3 is >4.2 ≤5 m/s; TD4 >4.2 ≤5 m/s; TD5>5 m/s; TD TD = Total Distance, meters per second, m/s; DEC = Deceleration, number, *n*; ACC = Acceleration, number, *n*; PL (Player Load, arbitrary unit, AU; MRS (Maximum Running Speed, meters per second m/s; W:R (Work: Rest Ratio, arbitrary unit, AU; EI (Exertion Index, arbitrary unit, AU. * A criterion of > 0.7 was considered to identify substantial loadings on each factor.

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
