# Peer review of "Comparison of the Physical Demands of Friendly Matches and Different Types On-Field Integrated Training Sessions in Professional Soccer Players"

_ijerph, 2020, doi:10.3390/ijerph17082904_

Round 1

Reviewer 1 Report

The study compares the physical demands of TS and FM. I'm not an expert in soccer so my comments are mainly methodological. The number of subjects is very small and I cannot trust the power of the study, but given the real world constraints imposed on such a study, let's proceed. Several suggestions. The first table of results (table 2) is overloaded. Try to provide the reader with a simple and general answer to the main research question, maybe by using MANOVA, and only than get into the details. Second, given the small number of participants consider the use of Bayesian Methods maybe by using the free software JASP. Consider the use of measures such as the Bayes Factor and not only the use of statistical significance tests, that have been severally criticized in almost any respected scientific field. A paper measuring the effect of some intervention and relying on the use of statistical significance tests and p-values is probably meaningless. In sum, present a comprehensive result summarizing your main research question. Use other methods (e.g. Bayesian or Machine Learning algorithms as they appear in JASP for instance), in order to convince the results in the validity of your findings. 

Author Response

ID: ijerph-772919- Minor Revisions

Title: Comparison of the physical demands of friendly matches and different

types on-field integrated training sessions in professional soccer players.

Dear Editor,

Thank you for the opportunity to re-submit the manuscript after revisions. We have followed the suggestions from reviewers (please see below our detailed responses to their comments).

Comments from reviewers:

Reviewer #1:

General Comments

REV1.- The study compares the physical demands of TS and FM. I'm not an expert in soccer so my comments are mainly methodological. The number of subjects is very small and I cannot trust the power of the study, but given the real world constraints imposed on such a study, let's proceed. Several suggestions.

AUT.- Thanks for your review and constructive comments.

REV1.- The first table of results (table 2) is overloaded. Try to provide the reader with a simple and general answer to the main research question, maybe by using MANOVA, and only than get into the details.

AUT.-  Thanks for you suggestion, as we have p-values and ES included into test we have deleted those columns of pairwise comparisons in table 2. Then, we tried to improve clarity of results presentation.

REV1.- Second, given the small number of participants consider the use of Bayesian Methods maybe by using the free software JASP. Consider the use of measures such as the Bayes Factor and not only the use of statistical significance tests, that have been severally criticized in almost any respected scientific field.

AUT.-  Thanks for this suggestion. We consider that the use of Bayesian models are of importance to Sport Sciences area due to the powerful findings when using small samples. However, the main aim of the current manuscript was to identify the dimensions of physical performance and their relationships among the different training tasks. In fact, we run the one-way ANOVA in order to check significant differences among tasks, and then, to justify the importance of PCA (unsupervised linear dimensionality reduction algorithm) to find a more meaningful basis or coordinate system from our data (finding the strongest features for each training task). Therefore, we followed the statistical procedures of previous studies to justify its use:

6.- Zurutuza, U., Castellano, J., Echeazarra, I., Guridi, I., & Casamichana, D. (2019). Selecting training-load measures to explain variability in football training games. Frontiers in Psychology, 10.

7.- Svilar, L., Castellano, J., Jukic, I., & Casamichana, D. (2018). Positional differences in elite basketball: selecting appropriate training-load measures. International journal of sports physiology and performance, 13(7), 947-952.

8.- Castellano, J., & Pic, M. (2019). Identification and Preference of Game Styles in LaLiga Associated with Match Outcomes. International Journal of Environmental Research and Public Health, 16(24), 5090.

In addition, as the suggestion is really useful we amend the use of Bayesian models in limitations of the study and further research paragraph in order to include in future studies these analyses with small sample sizes.

 REV1.- A paper measuring the effect of some intervention and relying on the use of statistical significance tests and p-values is probably meaningless.

AUT.-  Thanks for your suggestion. Please see our response to previous comment.

REV1.- In sum, present a comprehensive result summarizing your main research question.

AUT.-  Thank you for your suggestion. The author in line 325-332 (summarizing the main finding of current research).

REV1.- Use other methods (e.g. Bayesian or Machine Learning algorithms as they appear in JASP for instance), in order to convince the results in the validity of your findings. 

AUT.-  Thank you for you suggestion. In particular, we have followed the machine learning models established by Robertson (2015) with the use of PCA with continuous variables (physical performances) as an unsupervised model of machine learning (Robertson, S. (2015). Games by numbers: machine learning is changing sport. Retrieved from: http://theconversation.com/games-by-numbers-machine-learning-is-changing-sport-38973 (28-09-2016). Then, we consider the Bayesian models for further research when comparing or testing groups differences or tasks differences. Accordingly, we have added the cited references and one limitation/further research issue based on this recommendation.

Thanks for your constructive review and useful suggestions.

The authors

Reviewer 2 Report

Thank you for the opportunity to review your manuscript 'Comparison of the physical demands of friendly matches and different types on-field integrated training sessions in professional soccer players'. I do have a few comments for the authors to consider:

Keywords. Please, use MeSH terms (https://www.ncbi.nlm.nih.gov/mesh/). The MeSH terms will help to categorize and find your manuscript if it is published. For example, soccer is a MeSH term.

Material and Methods. Authors must follow a logical order: design, setting, participants ...

What was the study design?

How did you select the participants? Was it a convenience sample?

In the participants section you have included the sociodemographic results. This part should be included in the results section.

In the experimental design you have included the inclusion criteria of the participants, this part must be specified there.

The study was approved by the university ethics committee of the Faculty of Physical Activity and Sport Sciences, please include the assigned code.

The authors indicate that a 5-week follow-up was performed. On what date (month-year) did this follow-up take place?

Results. You can start this section with the sociodemographic data of the participants.

Author Response

ID: ijerph-772919- Minor Revisions

Title: Comparison of the physical demands of friendly matches and different

types on-field integrated training sessions in professional soccer players.

Dear Editor,

Thank you for the opportunity to re-submit the manuscript after revisions. We have followed the suggestions from reviewers (please see below our detailed responses to their comments).

Comments from reviewers:

Reviewer #2:

Suggestions for Authors

REV2.- Thank you for the opportunity to review your manuscript 'Comparison of the physical demands of friendly matches and different types on-field integrated training sessions in professional soccer players'. I do have a few comments for the authors to consider: Keywords. Please, use MeSH terms (https://www.ncbi.nlm.nih.gov/mesh/). The MeSH terms will help to categorize and find your manuscript if it is published. For example, soccer is a MeSH term.

AUT.- Thank you very much for your suggestion. We have changed the key words according to your recommendations.

REV2.- Material and Methods. Authors must follow a logical order: design, setting, participants.

AUT.- Thanks for this suggestion, the information was clarified accordingly in the Methods Subheadings section.

REV2.-What was the study design?

AUT.- The experimental randomized controlled trial was used (See, Experimental Design section you could check it, line 97)

REV2.- How did you select the participants? Was it a convenience sample?

AUT.-  Yes, the study sample was a convenience sample. We have clarified the section 2.2.

REV2.- In the participants section you have included the sociodemographic results. This part should be included in the results section.

AUT.- Thank you for your valuable appreciation, we will consider it for future studies. In the present research, the authors are not using sociodemographic variables. The section only includes the subjects’ age, height and body mass with the criterium of summarize the profile of sample.

REV2.- In the experimental design you have included the inclusion criteria of the participants, this part must be specified there.

AUT.- Thanks for your comment. The participant section has been rewritten.

REV2.- The study was approved by the university ethics committee of the Faculty of Physical Activity and Sport Sciences, please include the assigned code.

AUT.- The code has been inserted in participant section.

REV2.- The authors indicated that a 5-week follow-up was performed. On what date (month-year) did this follow-up take place?

AUT2.- The authors have rewritten the Experimental Design section

REV2.- Results. You can start this section with the sociodemographic data of the participants.

AUT.-   Thanks for your comment. However, as we did not analyzed any sociodemographic variable we cannot include those results here. Sorry for this misunderstanding.

Thanks for your constructive review and useful suggestions.